Multiple-file vs. single-file endodontics in dental practice: a study in routine care

Bartols Andreas andreas_bartols@azfk.de 1 2
Laux Gunter 3
Walther Winfried 1
1 Dental Academy for Continuing Professional Development Karlsruhe , Karlsruhe , Germany
2 Clinic for Conservative Dentistry and Periodontology, Christian-Albrechts-University Kiel , Kiel , Germany
3 University Hospital Heidelberg, Department of General Practice and Health Services Research, University of Heidelberg , Heidelberg , Baden-Württemberg , Germany
Druzinsky Robert
Electronic publication date: 2016 Dec 7
Publication date: 2016
Volume: 4
Electronic Location ID: e2765
Received 2016 Jul 13; Accepted 2016 Nov 6
Copyright: ©2016 Bartols et al.
Copyright year: 2016
Copyright holder: Bartols et al.
License: This is an open access article distributed under the terms of the Creative Commons Attribution License, which permits unrestricted use, distribution, reproduction and adaptation in any medium and for any purpose provided that it is properly attributed. For attribution, the original author(s), title, publication source (PeerJ) and either DOI or URL of the article must be cited.
License URL: https://creativecommons.org/licenses/by/4.0/

Keywords: Endodontics, Health services research, Patient outcomes, Dental public health, WaveOne, Single-file endodontics, Multiple-file systems, BioRaCe, Mity Roto Files, Clinical trial

Funding: “Health Services Research Baden-Württemberg” of the Ministry of Science, Research and Arts Social Order, Family, Women and Senior Citizens, Baden-Württemberg, Germany Dentsply Maillefer Andreas Bartols was funded within the young scientists’ program of the German network “Health Services Research Baden-Württemberg” of the Ministry of Science, Research and Arts, in collaboration with the Ministry of Employment and Social Order, Family, Women and Senior Citizens, Baden-Württemberg, Germany. Dentsply Maillefer made WaveOne motors available on loan and WaveOne files free of charge for the purpose of this study. The funders had no role in study design, data collection and analysis, decision to publish, or preparation of the manuscript.

==============================
Background

Little is known about the differences of rotary multiple file endodontic therapy and single-file reciprocating endodontic treatment under routine care conditions in dental practice. This multicenter study was performed to compare the outcome of multiple-file (MF) and single-file (SF) systems for primary root canal treatment under conditions of general dental practice regarding reduction of pain with a visual analogue scale (VAS 100), improvement of oral-health-related quality of life (OHRQoL) with the german short version of the oral health impact profile (OHIP-G-14) and the speed of root canal preparation.

Materials and Methods

Ten general dental practitioners (GDPs) participated in the study as practitioner-investigators (PI). In the first five-month period of the study, the GDPs treated patients with MF systems. After that, the GDPs treated the patients in the second five-month period with a SF system (WaveOne). The GDPs documented the clinical findings at the beginning and on completion of treatment. The patients documented their pain and OHRQoL before the beginning and before completion of treatment.

Results

A total of 599 patients were included in the evaluation. 280 patients were in the MF group, 319 were in the SF WaveOne group. In terms of pain reduction and improvement in OHIP-G-14, the improvement in both study groups (MF and SF) was very similar based on univariate analysis methods. Pain reduction was 34.4 (SD 33.7) VAS (MF) vs. 35.0 (SD 35.4) VAS (SF) (p = 0.840) and the improvement in OHIP-G-14 score was 9.4 (SD 10.3) (MF) vs. 8.5 (SD 10.2) (SF) (p = 0.365). The treatment time per root canal was 238.9 s (SD 206.2 s) (MF) vs. 146.8 sec. (SD 452.8 sec) (SF) (p = 0.003).

Discussion

Regarding improvement of endodontic pain and OHRQoL measure with OHIP-G-14, there were no statistical significant differences between the SF und the MF systems. WaveOne-prepared root canals significantly faster than MF systems.

Introduction

Clinical endodontic research is mainly conducted by specialists or specialized university centers (Friedman, Furberg & DeMets, 2010; Ng et al., 2007). The predominant types of such studies are retrospective observational studies, prospective cohort studies and a few randomized controlled trials (RCTs) (Ng et al., 2007). On account of the controlled study design, these studies have greater internal evidence and are classified as efficacy studies (Pfaff, Nellessen-Martens & Scriba, 2011). The effectiveness of endodontic interventions under everyday general dental care conditions has so far been hardly investigated (Nixdorf et al., 2012). Yet, patients treated in specialized centers can differ systematically from patients treated in routine care (Hulley, 2013).

A commonality of many experimental endodontic studies is the low number of cases (Peters & Wesselink, 2002; Pettiette, Delano & Trope, 2001; Weiger, Rosendahl & Lost, 2000). Larger case numbers are described for retrospective observational studies and prospective cohort studies, which, however, are often conducted without controls (Ng et al., 2007). Convincing results though can be obtained in studies if they include an adequate number of cases (Hulley, 2013). Since only few experimental endodontic studies have been made and many of them are lacking sufficient patient numbers, one could assume this to be an indication of considerable feasibility problems of such studies.

Reciprocating single-file (SF) systems are the latest stage of development of nickel-titanium (NiTi) instruments for the preparation of root canals (Bürklein, Benten & Schäfer, 2013; Yared, 2008). During the last years several systems as Reciproc (VDW, Munich, Germany), WaveOne (Dentsply, Konstanz, Germany), Genius files (Ultradent, South Jordan, UT, USA) or the Twisted Files Adaptive System (Kerr, Orange, CA, USA) with a combination of rotary and reciprocating movement were introduced into the market. Our knowledge of the clinical effects of using different systems for root canal preparation is limited (Schäfer, Schulz-Bongert & Tulus, 2004). The Swedish Council on Health Technology Assessment stated in its Systematic Review of Methods of Diagnosis and Treatment in Endodontics that the use of new tools facilitates the technical procedures of root canal treatment and that therefore investigations are needed regarding what influence these techniques have on everyday general practice (Bergenholtz et al., 2012).

Typically, new instrument systems are investigated in in-vitro-studies with extracted teeth (Bürklein, Benten & Schäfer, 2013) or root canal models (Goldberg, Dahan & Machtou, 2012). In such studies, the outcomes are mainly surrogate parameters, such as root canal straightening, preparation faults, preparation time in a workbench situation etc. the clinical significance of which can only be estimated to a limited extent (Hülsmann, 2013). Most of the few clinical trials available investigated only one instrument system (Fleming et al., 2010; Su, Wang & Ye, 2011) and rarely allow a comparison with other instrument systems (Schäfer, Schulz-Bongert & Tulus, 2004). Recently some randomized controlled trials were published, that investigated single and multiple file systems for endodontic treatment regarding pain reduction after treatment and improvement in quality of life (Kherlakian et al., 2016; Pasqualini et al., 2016; Relvas et al., 2016). It is unclear if there exists an effectiveness-gap (Pfaff, Nellessen-Martens & Scriba, 2011) between the results of these controlled studies under the optimal treatment conditions of specialized treatment providers and the use of rotary multiple-file (MF) and SF systems in general dental practice.

Therefore, research is needed when new endodontic techniques are introduced into dental practice. The study we performed investigates the effects of these endodontic techniques on dental practice. For this purpose, it uses the methods of health services research which studies care processes under everyday conditions of dental practice (Pfaff et al., 2009). Short-term patient-relevant outcomes were in the center of the study.

The design we chose was a multicenter study in routine care. We started by evaluating the outcome of endodontic treatment using conventional MF instrument systems for root canal preparation. Then, the practitioner-investigators (PIs) were trained in single-file (SF) endodontics (WaveOne-Instruments, Dentsply Maillefer, Ballaigues, Switzerland). Subsequently we evaluated the outcome of endodontic treatments using WaveOne.

The following research hypotheses were investigated in our study:

Primary outcome criterion

Does root canal preparation using SF root canal instruments lead to more or less reduction of patients’ endodontic pain compared to using rotary MF instrument systems?

Secondary outcome criterion

Does root canal preparation using SF root canal instruments lead to more or less reduction of patients’ oral-health-related quality of life compared to using rotary MF instrument systems?

Tertiary outcome criterion

Does root canal preparation using a single-file system require less time compared to the MF systems?

Methods

Study design

We performed the present study as a multicenter clinical study. For the purpose of this study we formed a network of ten general dental practitioners (GDPs). They acted as PIs. We conducted the study in two phases (Fig. 1). In the first 5-month phase the GDPs performed the endodontic therapy with different rotary nickel-titanium (NiTi) MF systems (Table 1). Subsequently the GDPs were trained for the use of the WaveOne SF system (Maillefer, Ballaigues, Switzerland). In the second 5-month phase the PIs treated the patients with SF WaveOne instruments exclusively. After each 5-month phase there was a 3-month follow-up so that treatments started could be completed.

Figure 1 Project plan of the study.

During inclusion, phases patients were assigned to the different study groups. In the follow-up phases, the started treatments were finished (MF, multiple-file system; SF, single-file system WaveOne).

The authors of this study acted solely as investigators and did not treat patients.

The study was conducted in conformity with the Declaration of Helsinki and the Professional Code for Physicians of the Medical Council of the State of Baden-Württemberg. The Ethics Committee of the Baden-Württemberg Medical Council reviewed the study and approved it (AZ: F-2011-034-z).

Participants

Patient eligibility and recruitment

All patients of the ten PIs who required endodontic therapy were consecutively assessed for eligibility.

The following inclusion criteria were defined: patients had to be at least 18 years old and in need of initial orthograde root canal treatment.

Table 1 Number of patients recruited by practitioner-investigator (P-I) and study group distribution.

P-I ID	Instrument type	Total	
	MF	Target	Actual	SF	Target	Actual	Target	Actual	
1	BioRaCe	28	28	WaveOne	28	28	56	56	
2	RaCe	28	45	WaveOne	28	56	56	101	
3	BioRaCe	28	30	WaveOne	28	23	56	53	
4	BioRaCe	28	22	WaveOne	28	23	56	45	
5	Alpha Kite	28	38	WaveOne	28	32	56	70	
6	BioRaCe	28	26	WaveOne	28	20	56	46	
7	Mity Roto Files	28	49	WaveOne	28	95	56	144	
8	BioRaCe	28	15	WaveOne	28	4	56	19	
9	BioRaCe	28	10	WaveOne	28	17	56	27	
10	BioRaCe	28	17	WaveOne	28	21	56	38	
Total N		280	280		280	319	560	599	
Notes.

MF multiple-file system

SF single-file system (WaveOne)

The following exclusion criteria were defined: patients with hopeless teeth for periodontal or restorative reasons, patients treated for emergency reasons only, more than one symptomatic tooth requiring endodontic treatment at the same time in one patient, patients with other oral findings causing pain, patients with craniomandibular dysfunction and communication difficulties (e.g., patients were not able to read, understand and complete the study questionnaires in German language).

All patients were recorded by the assistant staff of the dental practice and asked for the reason if they refused to participate. Every patient was given the study education and information documents (informed consent) that had to be signed and submitted by the patient before the patient was included in the study.

Practitioner investigators (PIs)

The ten dentists who participated in the study were general dental practitioners with at least two years of professional experience in a general dental practice and without endodontic specialization. All participating dentists worked under the conditions of the German “Statutory Health Insurance.” The PIs were chosen as a convenient sample of dentists that wanted to change their endodontic treatment to single file systems within the next 6–12 months anyway. All practices were located in southwest Germany.

All PIs were familiar with root canal preparation using rotary NiTi instruments (Table 1) and used them routinely in their practice. All dentists followed the “Good Clinical Practice: Root Canal Treatment” Guideline of DGZMK (German Society of Dental, Oral and Craniomandibular Sciences) (Hülsmann & Schäfer, 2005) which contains essential key points of the Quality guidelines for endodontic treatment of the European Society of Endodontology (2006).

Study initiation at the PIs

Before the study started, all participating PIs were visited by the principal investigator (AB) in their practice. The dentists were informed about the object and purpose of the study and its practical implementation. Each PI was given a copy of the study protocol and all other study relevant files. The dentists were informed about the planned procedure with regard to patient recruitment, education/information and treatment.

Interventions

In the first phase of the study, from 09/2011 to 02/2012, all endodontic treatments were performed with rotary NiTi MF systems (Fig. 1). All MF systems were used according to the manufacturer’s instructions. In 03/2012 the PIs were trained for the SF system. The training course explained the theoretical bases of the WaveOne System (Maillefer, Ballaigues, Switzerland) and provided hands-on training on extracted teeth. After this one-day training course every participating dentist was able to prepare root canals by the new method in a reliable way. The training was followed by a two-week implementation phase in all participating dental offices. During that time, the PIs should learn to treat patients with the new instruments and gain experience. In case of difficulties, this procedure offered the chance of clarifying problems. In the second phase of the trial, from 04/2012 to 08/2012, all endodontic treatments were performed with the SF system. All other variables of the practice setting and the treatment procedures remained unchanged.

Before treatment, the affected tooth was anesthetized by local anesthesia. After local anesthesia the endodontic access cavity was prepared. All teeth were isolated with a rubberdam. Root canals were probed with K-steel files of ISO sizes 06, 08, 10 and 15, in order to create a glidepath up to ISO 15 throughout all phases of the study. The working length was determined electrometrically and/or by X-ray. The dentists prepared the root canals according to the details provided by the manufacturers of the different rotary preparation systems. In the second study phase, the root canals were prepared with the WaveOne instruments according to the manufacturer’s instructions. If the dentists needed an apical preparation size that is not included in the WaveOne System, the last ISO size was followed up by a single hand instrument of the desired size. During rotary or reciprocating preparation the root canals were rinsed with 1–3% NaOCl between every rotary instrument or in case of the SF system between every 3–4 picks with the WaveOne file. After complete preparation of the root canals they were irrigated with a final irrigation of NaOCl 1–3% and a calcium hydroxide dressing or the root canal filling was placed. After that the tooth was sealed provisionally bacteria-proof with a temporary bacteria tight seal. In the last appointment the root canal filling was placed or in case of single-visit endodontics a definitive coronal filling was applied.

Outcomes

The primary outcome of reduction of endodontic pain and the secondary outcome of improvement of oral-health-related quality of life was measured with a patient questionnaire. The questionnaire assessed the pain by the Visual Analog Scale (VAS 100) (Turk, 2011) and the oral-health-related quality of life with the items of the short version of the oral health impact profile (OHIP-G-14) (John, Micheelis & Biffar, 2004) wich is the German translation of OHIP-14 (Slade, 1997). The patients were asked about the biggest complaints (consisting of the VAS 100 and OHIP-G14) without pain medication in the week before treatment and in the week before completion of treatment. This was two weeks after initial treatment and in connection with either the placement of the root canal filling or the definitive coronal filling of the tooth. The questionnaires were filled in by the patients before treatment started or while local anesthesia was taking effect. Any patient questions were answered by the dental team.

The time needed for root canal preparation was measured by the dental assistant staff. The measurement started when the first rotating or reciprocating instrument was placed in the root canal and ended when the last instrument was removed. The root canal recapitulations during preparation and the irrigations were included in the time measurement. Probing and glidepath preparation before using the rotary instruments were not included in the time measurement. Nor were the final irrigation of the root canals and the placement of a dressing included. When a tooth had several root canals, the total preparation time of all canals was measured and divided by the number of root canals in order to determine the preparation time per canal.

Further questionnaires and data collection

In the PI questionnaires the clinical findings (dental chart, sensitivity test, percussion test, apical pressure point, periodontal probing depth, radiographic presence of apical periodontitis, number of prepared root canals, presence of fistulae), the time needed for root canal preparation, instrument fractures and procedural events were documented. In addition, a consecutive patient log was introduced to record, if possible, the patient’s reason for rejecting participation. The patient forms included the patient’s informed consent to participate in the trial, a questionnaire asking for demographic and basic medical data, and the above described pain questionnaire which consisted of the VAS and the Items of OHIP-G-14.

All patients that qualified for participation in the study were informed about the study by the PI personally. All PI forms were filled in by the assistant dental staff. The patient questionnaires were filled in by the patients themselves. All patient questionnaires were pseudonymized and collected in a sealed box. The PI forms were pseudonymized in the same way to be able to match the patient data and the PI data in the subsequent evaluation.

The pain questionnaires were completed by the patients immediately before treatment started. The demographic data could be provided at any time, but were requested on completion of the treatment at the latest. The pain questionnaires were completed by the patients again 14 days after treatment. The PIs’ treatment was taken down on record. The time required for root canal preparation was measured by the dental assistant staff.

The questionnaires were handed over to the principal investigator (AB) at the end of the first and at the end of the second trial phase for evaluation.

Safety measures

Before treatment started, each patient participating in the study was informed about the endodontic risks in the same way as it is usually done before endodontic therapy. The patient was informed in particular about events, such as instrument fractures and other complications that may occur during root canal preparation and could lead to the extraction of the tooth affected. The patient was also informed, that root canal treatment is the last attempt to save a tooth. The information was provided by the PI and an additional education and information questionnaire. If in the course of the trial the complication of an instrument fracture occurred, the patient would be informed about it. This information was provided by an information questionnaire for instrument fractures.

Sample size calculation and statistics

For sample size calculation we had to consider the sample design which was characterized by a 2-stage structure (dental practice, patient). This cluster sample made special demands on both sample size planning and the analysis of the results (Donner & Klar, 2000).

To calculate the case number, the following parameters were defined: Significance level: 0.05, Power: 80% and Number of clusters (dental practices): 10.

Moreover, based on the analysis of similar studies (Pak, 2012), the most realistic assumptions possible were made about the clinically relevant difference of the VAS 100 (Visual Analog Scale) and the ICC (Intra Cluster Correlation Coefficient) which is a measure for the homogeneity in relation to a target variable of interest within the cluster: ΔV AS = 20 (20%) and ICC = 0.04.

For case number calculation, a validated software tool was used which determined the number of trial units (patients) per cluster (practice) on the basis of the parameters specified above (Campbell et al., 2004).

The resulting number of patients was 28 per dental practice and every trial phase. This number appeared realistic in terms of feasibility. In view of the basic statistical data on dental care in Germany (KZBV, 2011), a conservative estimate showed that one GDP performs about 60 root canal treatments per year. This means that 10 participating GDPs should be able to recruit the required case number in each of the trial phases.

The results were calculated with the SPSS (Version 21, Win x64) statistical system and SAS (Version 9.2, Win x64). With the PROC MIXED procedure (Singer, 1998) SAS offers options for explicitly considering potential cluster effects (here: several data collection units per dental office) in the overall regression model.

Assessment of potential covariates

Besides collecting the data for the primary outcome we assessed other dentist- and patient-related as well as treatment-specific covariates. This was done with a questionnaire for demographic information which also documented the patients’ basic medical data. In addition, the PIs recorded the dental chart and treatment-specific findings (tooth sensitivity before treatment, apical translucency, percussion test, apical pressure point and fistula).

Study termination criteria

It was planned to terminate the study when two weeks after root canal preparation by the new single-file method the patient’s pain was 40% above the expected level. The study would also have been terminated if single-file endodontics would have caused markedly more instrument fractures than expected. If during the study more than three instrument fractures had already occurred in the first 20 single-file treatment cases, the study would have been terminated.

Figure 2 Flow diagram.

Results

The ten PIs screened a total of 668 patients who met the study inclusion criteria. Of the 668 patients screened, 62 (9.2%) could not be included in the study primarily. In the course of the study, seven (1.0%) patients did not keep the agreed appointments for starting the treatment (Fig. 2). The remaining 599 patients were included into the study for statistical analyses. Thus, the number of patients actually included was 10.3% higher than the minimum required case number of 560 determined by power analysis.

Table 1 shows the number of patients that were recruited by the PIs. During both trial periods the individual participating GDPs recruited between 19 (PI 8) and 144 (PI 7) patients. All GDPs together included 280 (46.7%) patients for MF treatment and 319 (53.3%) patients for SF treatment. The distribution between the groups was nearly equal. The average age of the patients was 50.2 (SD 15.7) years. The distribution of the patients to the various participating GDPs differed (χ2; P < 0.001) whereas the age distribution across the study groups (MF and SF) showed no statistical differences (T-Test; P = 0.991) and was similar in both groups, i.e., 50.1 (SD 15.0) years in MF and 50.2 (SD 16.4) in SF. More men (53.1%) were treated than women. The gender distribution showed no statistical differences in the individual dental practices (χ2; P = 0.082) nor in the study groups (χ2; P = 0.458). The various types of vocational qualification, as stated by the study participants, differed in the individual practices (χ2; P < 0.001) but not in the study groups (χ2; P = 0.102) (Table 2).

The return rates of the various questionnaires that had to be completed by the patients and the participating GDPs were between 86% for the follow-up patient pain questionnaire and up to 97% for the questionnaire to be completed by the GDPs. The return rate of the patient pain questionnaire before treatment was 94% and for the demographic data questionnaire it was 90% (Table 3).

Table 2 Important socio-demographic characteristics and DMF-T of study participants by study center and study group.

Socio-demographic data of participants	Total	Study center	Study group	
		1	2	3	4	5	6	7	8	9	10	MF	SF	
Age (yr)														
Mean	50	52	46	55	53	49	52	47	59	49	61	50	50	
SD	16	15	13	17	16	13	17	16	16	15	14	15	16	
Range	18–88	20–83	18–79	19–88	19–79	23–85	23–84	19–80	35–84	19–74	29–79	18–85	19–88	
Total N	548	55	96	50	45	48	37	141	18	25	33	245	303	
Gender (female)														
N (%)	269	28 (51)	42 (42)	35 (66)	16 (36)	23 (43)	21 (48)	68 (48)	7 (37)	16 (59)	13 (37)	126 (49)	143 (46)	
Total N	573	55	101	53	44	53	44	142	19	27	35	259	314	
DMF-T														
Mean	12.0	15.5	7.7	16.2	17.0	7.8	11.4	9.3	17.4	10.0	20.7	12.0	12.0	
Total N	550	56	96	50	45	50	45	137	15	18	38	249	301	
Highest education														
Completed apprenticeship N (%)	241 (46)	31 (58)	40 (40)	15 (41)	22 (65)	17 (35)	15 (37)	57 (43)	13 (87)	20 (74)	11 (34)	95 (41)	146 (51)	
Technical/ Vocational school N (%)	97 (19)	10 (19)	12 (12)	12 (32)	7 (21)	13 (27)	9 (22)	20 (15)	1 (7)	4 (15)	9 (28)	52 (22)	45 (16)	
University/ College N (%)	92 (18)	7 (13)	33 (33)	1 (3)	0 (0)	10 (21)	12 (29)	17 (13)	1 (7)	0 (0)	11 (34)	47 (20)	45 (16)	
Other N (%)	40 (8)	1 (2)	9 (9)	3 (8)	1 (3)	4 (8)	4 (10)	17 (13)	0 (0)	1 (4)	0 (0)	19 (8)	21 (7)	
No N (%)	51 (10)	4 (8)	6 (6)	6 (16)	4 (12)	4 (8)	1 (2)	23 (17)	0 (0)	2 (7)	1 (3)	21 (9)	30 (11)	
Total N	521	53	100	37	34	48	41	134	15	27	32	234	287	
Notes.

MF multiple-file system

SF single-file WaveOne

Table 3 Questionnaire return rates for the enrolled participants.

Description	Timing	N (received)	N (expected)	%	
Patient survey demographic data	Before root filling	538	599	90	
Pain and OHIP-14 survey before treatment	1st appointment	565	599	94	
Dentist survey for treatment parameters	All appointments	582	599	97	
Patient 2 weeks follow-up survey	2 weeks after RCT	518	599	86	

In the course of the study slightly more maxillary (53.7%) teeth were treated (Table 4). The distribution of the different types of teeth showed no significant differences between the two study groups (χ2; P = 0.255).

Table 4 Descriptive data of teeth treated by location, study center (PI) and study group.

Tooth type	Study center	Study group	
		1	2	3	4	5	6	7	8	9	10	Multiple file	Single file	Total	
		Mean	(SD)	N	Mean	(SD)	N	Mean	(SD)	N	Mean	(SD)	N	Mean	(SD)	N	Mean	(SD)	N	Mean	(SD)	N	Mean	(SD)	N	Mean	(SD)	N	Mean	(SD)	N	Mean	(SD)	N	Mean	(SD)	N	Mean	(SD)	N	
Maxillary anteriors	No. of treated root canals	1.0	0.0	7	1.0	0.0	13	1.0	0.0	11	1.0	0.0	5	1.0	0.0	2	1.0	0.0	7	1.1	0.3	23	1.0	0.0	2	1.0	0.0	2	1.0	0.0	8	3.1	0.5	58	3.0	0.5	80	3.1	0.5	138	
	Time for treatment per tooth (s)	225.4	145.6	7	139.7	83.1	13	114.2	53.1	11	132.8	77.7	5	169.0	58.0	2	163.6	193.8	7	157.7	200.3	22	50.0	14.1	2	93.0	52.3	2	1500.0	2342.3	8	836.8	747.5	57	375.5	751.8	78	570.3	781.4	135	
	Time for treatment per root canal (s)	225.4	145.6	7	139.7	83.1	13	114.2	53.1	11	132.8	77.7	5	169.0	58.0	2	163.6	193.8	7	155.0	201.8	22	50.0	14.1	2	93.0	52.3	2	1500.0	2342.3	8	273.1	246.1	57	120.3	204.8	77	185.3	235.0	134	
Maxillary premolars	No. of treated root canals	1.6	0.5	11	1.4	0.5	17	1.7	0.5	7	2.0	0.0	9	2.0	0.0	5	1.7	0.5	6	1.8	0.4	20	2.0	0.0	2	1.6	0.5	8	1.3	0.5	7	1.7	0.5	53	1.6	0.5	39	1.7	0.5	92	
	Time for treatment per tooth (s)	276.6	115.0	11	188.1	104.8	17	274.7	116.9	7	150.6	42.6	10	322.2	45.8	5	165.5	84.4	6	299.2	271.9	19	87.5	17.7	2	249.1	150.0	8	950.0	435.0	6	303.0	200.2	53	257.0	326.1	38	283.8	259.6	91	
	Time for treatment per root canal (s)	180.0	75.0	11	141.6	69.4	17	171.1	105.5	7	74.3	22.4	9	161.1	22.9	5	97.6	31.2	6	162.1	130.8	19	43.8	8.8	2	148.2	70.0	8	710.0	210.1	6	192.5	154.3	52	164.4	197.8	38	180.7	173.5	90	
Maxillary molars	No. of treated root canals	3.2	0.4	13	2.9	0.4	26	3.1	0.7	11	3.0	0.7	9	3.3	0.5	14	3.1	0.4	8	3.1	0.3	37	2.3	0.6	3	3.2	0.8	6	3.0	0.4	11	1.0	0.0	37	1.1	0.3	43	1.0	0.2	80	
	Time for treatment per tooth (s)	908.6	1616.5	12	310.7	143.9	26	354.3	162.0	11	207.6	54.4	9	484.9	132.4	14	179.4	182.3	9	503.7	690.3	35	255.0	336.6	2	433.7	205.9	6	2100.0	523.8	11	274.9	229.6	37	294.8	1115.9	42	285.5	824.0	79	
	Time for treatment per root canal (s)	252.3	397.1	12	109.3	53.2	26	113.0	42.1	11	70.3	17.0	9	151.0	47.5	14	63.7	63.7	8	164.9	226.7	35	86.4	110.2	2	133.7	46.9	6	700.9	141.5	11	274.9	229.6	37	293.4	1116.2	42	284.7	824.2	79	
Mandibular anteriors	No. of treated root canals	1.0	0.0	1	1.0	0.0	3	1.0	0.0	7	1.0	0.0	3	1.0	0.0	2	1.0	0.0	2	1.1	0.4	14	1.0	0.0	3	1.0	0.0	2	1.0	0.0	2	3.0	0.6	61	3.0	0.6	78	3.0	0.6	139	
	Time for treatment per tooth (s)	114.0	0.0	1	122.7	76.2	3	113.7	38.4	7	55.0	35.5	3	229.5	113.8	2	50.0	31.1	2	271.1	268.9	14	45.0	5.0	3	172.0	1.4	2	1320.0	1018.2	2	669.7	662.4	62	278.0	324.5	78	451.5	537.6	140	
	Time for treatment per root canal (s)	114.0	0.0	1	122.7	76.2	3	113.7	38.4	7	55.0	35.5	3	229.5	113.8	2	50.0	31.1	2	249.3	259.6	14	45.0	5.0	3	172.0	1.4	2	1320.0	1018.2	2	219.6	205.0	60	93.6	107.3	77	148.8	169.1	137	
Mandibular premolars	No. of treated root canals	1.0	0.0	5	1.0	0.0	15	1.0	0.0	6	1.0	0.0	7	1.0	0.0	3	1.0	0.0	13	1.1	0.3	28	1.0	0.0	2	1.0	0.0	6	1.0	0.0	2	1.0	0.2	35	1.0	0.2	52	1.0	0.2	87	
	Time for treatment per tooth (s)	162.2	101.9	5	104.9	74.0	15	162.8	65.2	6	99.3	55.8	7	327.0	96.0	3	122.5	128.2	13	196.8	294.5	28	75.0	21.2	2	204.2	149.4	6	810.0	466.7	2	260.9	250.0	35	115.5	173.9	52	174.0	218.6	87	
	Time for treatment per root canal (s)	162.2	101.9	5	104.9	74.0	15	162.8	65.2	6	99.3	55.8	7	327.0	96.0	3	122.5	128.2	13	162.7	188.8	27	75.0	21.2	2	204.2	149.4	6	810.0	466.7	2	237.7	156.6	34	114.2	174.4	52	163.0	177.4	86	
Mandibular molars	No. of treated root canals	3.0	0.0	19	2.8	0.5	25	2.7	0.7	10	2.8	0.8	11	3.2	0.5	28	2.9	0.9	9	3.3	0.5	20	2.8	0.4	6	2.7	0.6	3	3.3	0.5	8	1.1	0.3	16	1.0	0.2	23	1.1	0.2	39	
	Time for treatment per tooth (s)	378.8	195.7	19	274.6	166.3	27	247.9	92.2	10	198.4	48.5	11	426.3	169.0	28	295.4	193.0	9	592.9	822.7	19	153.2	63.0	6	277.0	23.5	3	2040.0	521.1	8	276.1	239.8	16	195.6	421.0	23	228.6	356.2	39	
	Time for treatment per root canal (s)	126.3	65.2	19	98.6	57.3	25	96.6	42.4	10	71.8	13.1	11	135.3	53.1	28	93.7	53.8	9	181.9	263.5	19	53.9	22.2	5	108.6	33.4	3	635.6	168.4	8	259.2	229.2	16	194.1	421.5	23	220.8	353.1	39	
Totals	No. of treated root canals	2.3	1.0	56	2.0	1.0	99	1.9	1.0	52	2.1	1.0	44	2.8	0.9	54	1.8	1.0	45	2.0	1.0	142	1.9	0.9	18	1.9	1.0	27	2.1	1.1	38	2.1	1.0	260	2.1	1.0	315	2.1	1.0	575	
	Time for treatment per tooth (s)	429.9	788.1	55	222.2	148.5	101	217.8	134.9	52	157.3	68.9	45	409.5	160.7	54	176.2	164.8	46	345.7	524.3	137	117.0	114.5	17	266.0	170.6	27	1658.9	1218.4	37	496.2	555.8	260	268.9	607.3	311	372.4	594.8	571	
	Time for treatment per root canal (s)	180.2	200.7	55	115.9	66.0	99	123.8	61.7	52	82.2	41.3	44	157.1	69.2	54	106.1	109.7	45	173.5	211.9	136	57.2	34.8	16	150.7	86.3	27	900.4	1114.8	37	238.9	206.2	256	147.1	453.5	309	188.7	365.5	565	

Evaluation of the primary outcome criterion

For the evaluation of the primary outcome, i.e., post-operative reduction of patients’ endodontic pain and improvement of oral-health-related quality of life, we measured pain reduction via VAS 100 and the OHIP-G14 score. Both values were measured before root canal treatment and 14 days after treatment. Then we compared the different study groups (MF and SF).

The mean pain score before root canal treatment for MF was 42.3 (SD 32.6) VAS and for SF 43.9 (SD 32.0) VAS and decreased to 10.0 (SD 18.6) VAS (MF) and 9.3 (SD 19.2) VAS (SF).

The mean OHIP-G 14 score before root canal treatment for MF was 12.5 (SD 10.6) and for SF 13.0 (SD 10.8) and decreased to 3.6 (SD 5.1) (MF) and 4.6 (SD 6.5) (SF).

For pain reduction and OHRQoL, univariate analysis showed a very similar improvement in both study groups (MF and SF):

(a) Pain reduction 34.4 (SD 33.7) VAS (MF) vs. 35.0 (SD 35.4) VAS (SF) (p = 0.8).

(b) Improvement of oral-health-related quality of life according to OHIP-14 score: 9.4 (SD 10.3) (MF) vs. 8.5 (SD 10.2) (SF) (p = 0.4).

The differences between the study groups were not significant.

Multivariate analysis of variance (MANOVA) taking the additional factor “single vs. multiple-visit treatment” into account did not reveal any significant influence of the factors “study group (MF or SF)”, “single vs. multiple-visit” or an interaction of the two regarding pain reduction. For improvement of OHIP-14 score there was an overall significant influence (p = 0.03) with “single-visit” treatments having a significantly (p = 0.01) lower improvement than “multiple-visit”. But further analyses showed that OHIP-14 scores for “single-visit” treatments (10.7 (SD 9.9)) were already significantly (p = 0.06) lower before treatment than for “multiple-visit” treatments (13.6 (SD 10.9)) and dropped to almost the same levels before completion of treatment (4.1 (SD 5.9) vs. 4.2 (SD 6.0)).

Evaluation of the secondary outcome criterion

For the speed of root canal preparation, univariate analysis showed a significant difference between the study groups MF and SF: For (c) a multivariate analysis was made taking into consideration the dentist- and patient-related as well as treatment-specific covariates:

(c) Duration of treatment per root canal (in sec): 239 (MF) vs. 147 (SF) (P = 0.003).

• Gender of dentist

• Gender of patient

• Age of patient

• Comorbidities of the patient (none, hypertension, DM(I or II), asthma)

• DMFT Index

• Tooth

• Tooth sensitivity before treatment (positive/negative)

• Apical translucency (yes/no)

• Percussion test (positive/negative)

• Apical pressure point (yes/no)

• Fistula (yes/no).

For the WO SF system a significantly shorter duration resulted in comparison to the MF systems (121 s; SD 37.40; p = 0.01). The adjusted reduction in required preparation time was 32.8% with the SF-System.

The root canal preparation with the WaveOne System produced the same results regarding reduction of patients’ endodontic related pain and oral health related quality of life, but the preparation speed per root canal was faster.

Discussion

Our study showed that root canal treatment with MFs as well as with the SF WaveOne System reduced the patients’ endodontic related pain and improved oral health related quality of life without statistically significant differences under conditions of general dental practice. The root canal preparation with the SF system was faster.

Measuring patients’ endodontic pain and oral-health-related quality of life

Pain intensity can be measured with various methods, e.g., the Visual Analog Scale (VAS), the Numerical Rating Scale (NRS) or the Verbal (Categorical) Rating Scale (VRS). The VRS is an easy-to-apply measuring method, but forces the study subjects to select a wording which may not represent an adequate description of the pain they feel. Moreover, the VRS depends on a clear and unequivocal understanding of the language (Turk, 2011). The NRS and VAS are uncomplicated measuring methods for the pain felt and both show good evidence of construct validation (Turk, 2011). In the present study, the VAS was selected because it offers a large number of scores. This makes the VAS more sensitive to changes in the pain intensity felt than other scales offering fewer reply categories (Turk, 2011). In addition, the VAS is widely used in the endodontic literature (King et al., 2012; Martin-Gonzalez et al., 2012; Pak, 2012; Udoye & Jafarzadeh, 2011). The limitation to one scale is both feasible and adequate (Attar et al., 2008).

Pain assessment alone gives no information about the patients’ oral-health-related quality of life which represents a patient-relevant outcome (Pfaff et al., 2009). The only validated German-language measuring instrument for the oral-health-related quality of life is the OHIP (John et al., 2003). To limit the questionnaire to a practicable length, the OHIP-G-14 was used in our study (John, Micheelis & Biffar, 2004). In 2011, when the study was planned, there existed two endodontic studies which chose the OHIP-14 questionnaire for endodontic issues (Dugas et al., 2002; Gatten et al., 2011).

Description of the results

In the present study, we compared two clinical short-term outcome parameters of two basic methods for root canal preparation.

We defined the most important patient-relevant outcomes as reduction of pain by endodontic therapy and the improvement of oral-health-related quality of life. When we were planning this study in 2011, shortly after the introduction of reciprocating SF endodontic instruments (WaveOne and Reciproc), there was no information about the clinical performance of these instruments. The new system should have at least a similar or an enhanced clinical outcome compared to the conventional (MF) systems. This is an important condition when a new technology is introduced. Moreover, we expected greater speed and thus greater efficiency of the SF system as a relevant result. As far as the authors know, no studies had been made that compared the clinical outcome of MF and SF systems at the time of the study planning in 2011.

The reduction of pain and the improvement of oral health related quality of life as a result of endodontic treatment were not different in the two experimental groups (MF and SF) in the second week after treatment. Both methods were equally effective in reducing endodontic pain. The mean pain intensity of about 43.2 (SD 32.2) VAS before root canal treatment and about 9.5 (SD 19.0) VAS after treatment agrees with the results obtained by other researchers who investigated pain reduction after endodontic therapy (Ehrmann, Messer & Adams, 2003; Pak & White, 2011). A review found that pain levels before endodontic therapy of 54 VAS and a standard deviation of 24 VAS are a common average in endodontic studies and decrease to less than 10 VAS on average within 7 days (Ehrmann, Messer & Adams, 2003; Pak & White, 2011). Also Ehrmann, Messer & Adams (2003) found very similar values to those in our study (44.4 (SD 26.9) VAS). The mean pain measured 4 days after endodontic therapy decreased to 7.5 (SD 15.5) VAS. The mean pain reduction of 36.9 (SD 29.0) VAS was very similar to the values in our study. The improvement of the OHIP-14 with mean scores of 12.8 (SD 10.6) before therapy found in our study are very comparable to another study where a mean OHIP-14 score of 15.4 (SD 10.5) was found before endodontic treatment (Liu, McGrath & Cheung, 2014). Also other studies report that endodontic treatment leads to an improvement of the oral-health-related quality of life (Dugas et al., 2002; Hamasha & Hatiwsh, 2013). This finding was confirmed by our study.

Recently a couple of randomized controlled trials (RCT) have been published, that evaluated the pain reduction and/or the improvement in quality of life of single file systems (Kherlakian et al., 2016; Pasqualini et al., 2016; Relvas et al., 2016). In the first RCT (Kherlakian et al., 2016) two SF reciprocating systems (Reciproc (VDW, Munich, Germany) and WaveOne (Dentsply)) and one MF system (ProTaper Next (Dentsply)) were compared. Only asymptomatic vital teeth were treated with the different systems. Therefore patients did not have pain before treatment. Also after treatment, pain rates on a categorized VAS 100 score were also very low and showed no significant differences between the systems. The second RCT (Relvas et al., 2016) compared one reciprocating SF system (Reciproc (VDW, Munich, Germany)) with a MF system (ProTaper (Dentsply)). Only asymptomatic teeth with apical periodontitis were included in the trial. Therefore patients were pain-free before treatment. Pain measurement was not performed with the VAS. Therefore, results can be hardly compared with our study. The different instrument systems showed no statistically significant differences in postoperative pain scores after 24 h and 72 h. The third RCT (Pasqualini et al., 2016) investigated the ProTaper MF and the WaveOne SF system. Compared were primary root canal treatments of every clinical condition (symptomatic, asymptomatic, vital and non-vital cases). Mean pain on VAS was 35.2 for SF before treatment and 24.6 for MF decreasing to very low rates of 1.3 (SF) and 0.9 (MF) after seven days. This is different to our study, where the mean pain scores were higher before treatment, but more equal. Also, our mean pain scores in the second week after treatment were higher than the score found in the above mentioned study. If this can be interpreted as an effectiveness gap regarding the success of these instruments in general dental practice compared to specialized care providers remains unclear, because our initial VAS scores were higher and therefore perhaps naturally need a longer time period to drop to low scores. Recently our research group published a smaller, but very similar study comparing hand instrumentation with Reciproc for root canal preparation under conditions of general dental practice (Bartols et al., 2016). The mean pain score before root canal treatment with hand instruments was 43.6 (SD 30.7) VAS and with Reciproc it was 41.2 (SD 27.7) VAS, which is perfectly comparable with the initial values of the present study where the initial VAS scores were 42.3 (SD 32.6) VAS for MF and 43.9 (SD 32.0) VAS for SF. Within the same period of time the scores decreased in all four groups to values in the range between 9.3 and 11.5 (with SDs of 16.5–19.2) VAS therefore only showed minimal differences. Also, regarding OHRQoL, the OHIP-G-14 scores of all four groups show only minimal differences before treatment and before completion of therapy (hand instruments 9.2 (SD 9.6) decreasing to 3.4 (SD 5.4), Reciproc 10.4 (SD 9.6) decreasing to 3.5 (SD 6.1), MF 12.5 (SD 10.6) decreasing to 3.6 (SD 5.1) and SF WaveOne 13.0 (SD 10.8) decreasing to 4.6 (SD 6.5)). Therefore, it can be concluded that all four techniques investigated show the same clinical outcome regarding pain reduction and improvement in OHRQoL under routine care conditions.

As the main outcome parameter was the reduction of endodontic pain after treatment, the different root canal preparation techniques and their influence on postoperative pain has to be considered. Recently it was demonstrated that different root canal preparation techniques lead to the expression of different levels of inflammatory neuropeptides in the periapical periodontal ligament linked with the possible emergence of symptomatic apical periodontitis (Caviedes-Bucheli et al., 2013). It is believed that this is connected to the different amounts of extruded debris beyond the apical foramen (Caviedes-Bucheli et al., 2016). Since nearly all root canal instrumentation techniques including hand instrumentation as well as engine driven instruments lead to apical extrusion of debris (Al-Omari & Dummer, 1995; Bürklein & Schäfer, 2012; Capar et al., 2014; De-Deus et al., 2010) in most cases there will be an inflammatory response to a certain extent. In an in-vitro study reciprocating instruments extruded more debris than rotary instruments (Bürklein & Schäfer, 2012) with Reciproc producing most debris while another in-vitro study found Reciproc to produce significantly less extruded debris compared to rotary techniques (Kocak et al., 2013). The only clinical studies measuring the expression of inflammatory neuropeptides in the periodontal ligament found that the instrument design of engine driven root canal instruments has a greater impact on expression of neuropeptides than the instrumentation technique (Caviedes-Bucheli et al., 2016). Because of this contradictory data situation and the limited knowledge, if the amount of expressed neuropeptides can be directly correlated to the perceived pain it remains unclear if there is an impact on the postoperative pain levels of patients after root canal treatment. In our study the preoperative levels of pain, their improvement and the postoperative VAS pain levels were very similar and very much comparable to our previously published study (Bartols et al., 2016). Therefore in the heterogeneous situation of clinical cases, the impact of the root canal preparation systems used seems to be limited regarding postoperative pain relief.

A significant difference was found in the speed of root canal preparation. The preparation time required when using WO instruments was on average 92 s shorter than the time required with MF systems. This time is probably saved because the WO system does not require any instrument changes. As changing instruments cannot be avoided with MF systems, the time needed for it was included in the time measurement. An in-vitro study reported that root canal preparation with WaveOne instruments in contrast to MF systems is about 100 s faster (Bürklein et al., 2012). This time benefit per canal was also observed in our study. In both study designs the instrumentation time included instrument changes, cleaning of instruments and irrigation of the root canal. Therefore results are comparable. Thus, there is nearly no effectiveness gap of the method. This was not necessarily to be expected as unlike root canal preparation in the laboratory the preparation in the patient’s mouth is more complicated due to patient-related factors, such as mouth opening, restlessness of the patient etc. The time saved in canal preparation can be beneficially reinvested in additional root canal disinfection (Van der Sluis, Wu & Wesselink, 2009).

In general, the endodontic literature proves that pain that existed before endodontic therapy will be reduced by root canal therapy (Ehrmann, Messer & Adams, 2003; Genet, Wesselink & Thoden van Velzen, 1986; Pak & White, 2011). Comparative studies on endodontic pain have so far mainly compared different types of pain medication (Attar et al., 2008; Ryan et al., 2008), different types of root canal dressings (Ehrmann, Messer & Adams, 2003; Torabinejad et al., 1994) and differences between single-visit vs. multiple-visit treatment (Prashanth et al., 2011; Su, Wang & Ye, 2011). For single- versus multiple-visit treatment, studies found no differences for one week postoperative pain levels (Figini et al., 2008; Prashanth et al., 2011). This suits our results, because we also did not find differences in our analyses regarding single- versus multiple-visit treatments regarding pain reduction. For OHRQoL there was a difference in improvement of OHIP-14 scores between single- and multiple-visit treatments. But as the initial OHIP-14 scores were significantly lower in the single-visit group than the initial scores in the multiple-visit group, we conclude that the PIs primarily treated “safe” cases with low initial OHIP-14 scores as single-visit.

Clinical trials comparing pain after root canal preparation with different instrument systems are rare (Gambarini et al., 2013; Kherlakian et al., 2016; Pasqualini et al., 2016; Relvas et al., 2016) and have mostly low case numbers (N = 30–70 per experimental group). The authors do not know of any large-scale clinical comparative studies with high case numbers reliably reflecting the dental practice reality. Research in practice networks offers an environment which allows to generate case numbers high enough for clinical trials (Nixdorf et al., 2012). In this way, new research opportunities are created that can also be applied to other issues of endodontics or other fields of dentistry.

All three outcome parameters reflect patient relevant short term success criteria, that are not necessarily connected to the long term success of the treatments performed. To the knowledge of the authors, until now there is only one study that investigated also the long term success of technological change from stainless steel instrumentation to rotary instrumentation in a general dental practitioner situation in the Swedish Public Dental Service (Koch et al., 2015). While tooth survival was higher in teeth treated post-education with rotary instruments there was no improvement in periapical health. Only surrogate parameters like the technical quality of the treatments improved. Studies investigating the long term outcome after technological change to reciprocating technique in endodontics are not known to the authors. Therefore, further research regarding long term results of reciprocating techniques in general dental practice is needed.

Study design and feasibility

Generally this study was planned as a health services research study. Therefore, it was never intended to compare two treatment groups in a classical clinical trial setting. We wanted to investigate the effects of technological change in endodontic treatment in general dental practices and chose therefore a study design in a timeline sequence and not a study design with parallel treatments groups. We accompanied the technological change in endodontic treatment methods in everyday dental practice to uncover possible “shortcomings” or effectiveness gaps by structured observation, which were basically not found for short-term outcomes as pain reduction and improvement of OHRQoL.

In the present study, the number of recruited patients agreed with the initial case number planning. The planning therefore seemed to be based on realistic assumptions. The return rates of the collected study data and questionnaires were high and the patients were adequately followed up. Judged by these requirements, research can be conducted in dental practices in an adequate way (Kohout et al., 2015).

Ten GDPs agreed to participate in the study as PIs. This exactly equaled the number underlying the power analysis. The recruitment of the minimum number of required GDPs poses the risk that the case numbers aimed at cannot be reached. As studies of this type are rare in endodontics, there are no broadly-based typical figures available on the experience regarding the recruitment of PIs. There is only one study pursuing a similar approach by observing the results obtained in dental practices (Nixdorf et al., 2012). That study was designed as an observational study to measure pain and burden connected with initial orthograde root canal treatment. A total of 62 GDPs participated in the study, whereas 48 had been aimed for in case number planning. This corresponds to an over-recruitment rate of 29% (Nixdorf et al., 2012).

Contrary to that study (Nixdorf et al., 2012), the present study takes an approach to compare different treatment methods, which makes considerably higher demands on the participating GDPs. The GDPs had to undergo training to learn how to prepare the root canals with the SF WaveOne instruments and, at the same time, they had to care for two therapeutic groups and to recruit themselves the patients for each. The GDPs did not get any financial support. As an incentive, they were offered a payment of €5 for every evaluable/analyzable case which, however, most colleagues did not take. The training for the use of WaveOne instruments was provided free of charge to the GDPs. In addition, in the second study phase Dentsply Maillefer (Ballaigues, Switzerland) made available the required WaveOne files free of charge and loaned the GDPs the Wave-One motors. In view of the fact that the literature describes serious resentments of German physicians against practice-based clinical trials (Hummers-Pradier et al., 2012; Hummers-Pradier et al., 2008), it is a special success to recruit 10 GDPs. Moreover a recently published similar study of our research group showed, that 3 of 9 PIs could not cope with the organizational demands of a study like this and could not contribute any cases for evaluation (Bartols et al., 2016).

The participating GDPs documented treatments that were required anyway. The practice routine had to be changed for the documentation requirements of the study, but the organizational work with the study participants was mainly delegated to the assistant dental staff. This certainly is one reason for the good feasibility of the study. Moreover, there were no special demands on the patients, so that their willingness to participate in the study was very high. The GDPs screened 668 patients and actually enrolled 599 in the study so that on average every GDP screened about 1.1 patients to include one in the study. Compared with the study of Nixdorf et al. (2012) who screened 1.5 patients for each subject included in the study, this is a high rate of inclusion and shows the patients’ great willingness to participate in a clinical trial of the extent described here. Altogether 599 participants were recruited, whereas 560 would have been needed. This is an over-recruitment of not quite 7%, so that, on average, the case number aimed for was reached. However, the individual case numbers differed very much (Table 1).

The two-phase study design split into separate periods increased the GDPs’ willingness to participate in the clinical study because it limited the organizational effort for the study. Although this means that the present study was not based on randomization, generally considered the optimum study design (Friedman, Furberg & DeMets, 2010; Hulley, 2013), the clear time split of the study groups prevented randomization errors and selection bias at the level of the participating dental practices in the sense of manipulating the patient randomization to each of the study groups and was also used in another study investigating endodontic technological change in general dental practice (Koch et al., 2015). The consecutive sample used in the present study also counteracted the volunteer bias (volunteerism) (Hulley, 2013). Additionally the broad inclusion criteria for the participating patients made recruitment feasible for the PIs and reflects in this way the conditions of everyday general dental practice.

Conclusion

Concerning the reduction of endodontic pain and improvement of oral-health-related quality of life, the WaveOne SF system shows no statistical difference to MF systems under the conditions of general dental practice. The speed of preparation of root canals appears to be higher with the WaveOne SF instruments.

Supplemental Information

Data S1 Raw Data Supplemental File

Raw data collected from the patient questionnaires and the P-Is applied for preparation of Figs. 1–4.

Click here for additional data file.

The authors would like to thank all the participating dental practices and all patients for their cooperation. We are deeply grateful for the opportunity to conduct this study.

Additional Information and Declarations

Competing Interests

Author Contributions

Human Ethics

Data Availability

The authors declare there are no competing interests.

Andreas Bartols conceived and designed the experiments, performed the experiments, analyzed the data, contributed reagents/materials/analysis tools, wrote the paper, prepared figures and/or tables, reviewed drafts of the paper.

Gunter Laux and Winfried Walther conceived and designed the experiments, analyzed the data, contributed reagents/materials/analysis tools, wrote the paper, prepared figures and/or tables, reviewed drafts of the paper.

The following information was supplied relating to ethical approvals (i.e., approving body and any reference numbers):

The study was conducted in conformity with the Declaration of Helsinki and the Professional Code for Physicians of the Medical Council of the State of Baden-Württemberg. The Ethics Committee of the Baden-Württemberg Medical Council reviewed the study and approved it (AZ: F-2011-034-z).

The following information was supplied regarding data availability:

The raw data has been supplied as a Data S1.

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
