# Peer review of "Multiple-file vs. single-file endodontics in dental practice: a study in routine care"

_PeerJ, doi:10.7717/peerj.2765_

## Round 0.1 · original submission · Major Revisions

Please address all of the comments made by the reviewers in your revised manuscript.

Reviewer 1 ·

Basic reporting

Although this manuscript is well written, the experimental model is weak in regards to using a general dentist model only along with the experimental parameters used.

Experimental design

The model should have also include endodontists. This would better compare the outcomes of GPs to endodontists in regards to the experimental parameters tested. Also, pain and speed in regards to endodontic files does not necessarily have any correlation to long-term prognosis of an endodontically treated tooth. This is important to note since endodontic files clinically are selected for their efficiency and effectiveness in providing good long-term outcomes in conventional endodontic treatment and not reduction in pain.

Validity of the findings

In any clinical, multi-operator model, validity of results come into question. In example, the current model can not account for the pain that may come from factors other than file usage such as the chemo portion of the chemomechanical process of conventional root canal treatment.

Additional comments

Overall, although this manuscript is very detailed, a clinical file comparison model should be more directed towards outcomes of conventional endodontic treatment and not pain or speed of treatment. Because in clinical practice of endodontics, you can have no pain and uses files that are quick to perform endodontics, but the case can still fail long-term.

·

Basic reporting

ok

Experimental design

ok

Validity of the findings

ok

Additional comments

I have to make a compliment to the authors for this paper. being a practice based study with all related problems in design and getting reliable data, this study is an example for a thorough and well executed experiment. The outcome could be expected, which puts a question if all effort is worthwhile, but on the other hand, this paper can be used to show that the new technique has comparable outcome as the gold standard and has the additional advantage of a reduced treatment time. As the number of treatments is rather high, the resulting outcome shown in table 4 is interesting and could be made more interesting and relevant for the profession by making a graph or table showing for all teeth (molars/premolars/anterior teeth divided by jaw) the number of teeth, average number of treated canals by tooth group, average treatment time per tooth group and the average treatment times for practices / canal. The number of teeth and operators allows these additional data to be used in other studies as a reference for general data as how many endo's are done in front teeth, how many canals are found by GPs in teeth etc. It is just an advice, but i think the paper will gain relevance just by presenting the descritive data on all 583 endodontic treatments

---

## Round 0.2 · accepted · Accept

Thank you for submitting your revised manuscript.

·

Basic reporting

ok

Experimental design

ok

Validity of the findings

ok

Additional comments

acceptable for publication